# Bioprospecting of Rhizosphere-Resident Fungi: Their Role and Importance in Sustainable Agriculture

**DOI:** 10.3390/jof7040314

**Published:** 2021-04-18

**Authors:** Mahadevamurthy Murali, Banu Naziya, Mohammad Azam Ansari, Mohammad N. Alomary, Sami AlYahya, Ahmad Almatroudi, M. C. Thriveni, Hittanahallikoppal Gajendramurthy Gowtham, Sudarshana Brijesh Singh, Mohammed Aiyaz, Nataraj Kalegowda, Nanjaiah Lakshmidevi, Kestur Nagaraj Amruthesh

**Affiliations:** 1Applied Plant Pathology Laboratory, Department of Studies in Botany, University of Mysore, Manasagangotri, Mysuru 570006, Karnataka, India; botany.murali@gmail.com (M.M.); nazia@botany.uni-mysore.ac.in (B.N.); Knataraj922@gmail.com (N.K.); 2Department of Epidemic Disease Research, Institutes for Research and Medical Consultations (IRMC), Imam Abdulrahman Bin Faisal University, Dammam 31441, Saudi Arabia; maansari@iau.edu.sa; 3National Center for Biotechnology, Life Science and Environmental Research Institute, King Abdulaziz City for Science and Technology, Riyadh P.O. Box 6086, Saudi Arabia; malomary@kacst.edu.sa (M.N.A.); salyahya@kacst.edu.sa (S.A.); 4Department of Medical Laboratories, College of Applied Medical Sciences, Qassim University, Qassim 51431, Saudi Arabia; 5Central Sericultural Germplasm Resources Centre, Central Silk Board, Ministry of Textiles, Thally Road, TVS Nagar, Hosur 635109, Tamil Nadu, India; thrivenimc@gmail.com; 6Department of Studies in Biotechnology, University of Mysore, Manasagangotri, Mysuru 570006, Karnataka, India; gajendramurthygowtham@gmail.com (H.G.G.); brijeshrajput.bt@gmail.com (S.B.S.); reachaiyaz@gmail.com (M.A.); 7Department of Studies in Microbiology, University of Mysore, Manasagangotri, Mysuru 570006, Karnataka, India; lakshmiavina@rediffmail.com

**Keywords:** PGPF, plant growth, plant immunity, biotic stress, abiotic stress

## Abstract

Rhizosphere-resident fungi that are helpful to plants are generally termed as ‘plant growth promoting fungi’ (PGPF). These fungi are one of the chief sources of the biotic inducers known to give their host plants numerous advantages, and they play a vital role in sustainable agriculture. Today’s biggest challenge is to satisfy the rising demand for crop protection and crop yield without harming the natural ecosystem. Nowadays, PGPF has become an eco-friendly way to improve crop yield by enhancing seed germination, shoot and root growth, chlorophyll production, and fruit yield, etc., either directly or indirectly. The mode of action of these PGPF includes the solubilization and mineralization of the essential micro- and macronutrients needed by plants to regulate the balance for various plant processes. PGPF produce defense-related enzymes, defensive/volatile compounds, and phytohormones that control pathogenic microbes’ growth, thereby assisting the plants in facing various biotic and abiotic stresses. Therefore, this review presents a holistic view of PGPF as efficient natural biofertilizers to improve crop plants’ growth and resistance.

## 1. Introduction

The global human population is predicted to increase from the current day situation and is estimated to reach approximately 9.7 billion (i.e., a 24% increase) by the year 2050 [1]. Accordingly, food production should increase by above 70% of the current levels to feed the vast population. In order to supply food for the increasing population, pre- and post-harvest diseases must be controlled in order to secure food security. Apart from pathogens, the contamination caused by unnecessary agrochemicals and their misuse has also contributed to reduced production [2]. The findings have also prompted researchers to develop new approaches to the management of diseases caused by various agents, including pests. Rhizosphere-resident plant growth promoting fungi (PGPF) are one of the effective eco-friendly management strategies of plant diseases, and they may also serve as an alternative tactic to boost the growth and defense mechanism in plants. The application of PGPF will reduce the use of chemical control to a minimum, and it protects plants against various biotic and abiotic stresses. Many PGPF species (*Trichoderma*, *Talaromyces*, *Fusarium*, *Phytophthora*, *Penicillium*, *Rhizoctonia*, *Gliocladium*, *Phoma*, etc.) have been reported to date to boost plants’ growth, and to enhance their innate immunity and other important secondary metabolites in plants [3,4,5,6,7]. The above beneficial aspects rendered by PGPF are due to root colonization, the production of growth hormones, the solubilization of essential minerals, antagonistic properties, mycoparasitic and saprophytic resistance, and competition for space and nutrients, in addition to the induction of systemic resistance (ISR) in plants [8,9,10]. Apart from these, PGPF prevents pathogen infection in plants, improves soil nutrient availability, and reduces ethylene in the native host via the production of 1-aminocyclopropane-1-carboxylate (ACC) deaminase and phytohormones [7,11]. The overview of the beneficial importance of PGPF in the promotion of plant growth and resistance is represented in Figure 1 and Figure 2.

During ISR activation, PGPF treatment regulates the levels of defense signaling hormones such as jasmonic acid (JA) and/or ethylene (ET), along with the accumulation of certain proteins (thaumatin), and stimulates the production of defense-linked enzymes—chiefly GLU (glucanases) and CHI (chitinases)—that prevent the growth and proliferation of pathogens either directly or indirectly [12,13]. Before regulating defense signaling pathways, PGPF triggers the increased synthesis of secondary compounds (lignin, callose), along with the production of antioxidant enzymes that protect plants from various biotic and abiotic stresses [14,15]. Apart from the above-mentioned factors, the need for superior PGPF to reduce toxic chemical fertilizers has been highlighted as one of the key moves towards organic farming practices. Therefore, the review sheds light on the recent findings related to the application of rhizosphere-resident PGPF in improving plant growth, the elicitation of resistance, co-operation with the host, root colonization, and commercial formulation of byproducts (Figure 3). The information produced from this review could be of great benefit to those looking for an organic method of farming primarily to achieve sustainability of crop production.

## 2. Beneficial Aspects of Rhizosphere-Resident PGPF

### 2.1. Production of Phytohormones

Phytohormones play a crucial role in regulating plant growth and defense. Indole 3-acetic acid (IAA) is one of the most widespread natural auxins, which chiefly improve plants’ root growth and morphology [4,16]. Treatment with sterile PGPF—*Penicillium janczewskiy—*was reported to elicit IAA production, and it induced resistance against stem rot caused by *R. solani* by altering the root growth of melon plants. The involvement of auxin precursors in these phenomena has not been determined, but the improved root development is consistent with the observed effects of fungal-based auxin production [16]. Various species of *Trichoderma*, *Penicillium*, *Aspergillus*, *Fusarium*, *Talaromyces* and *Mortierella* isolated from the rhizosphere soils are reported to stimulate IAA production in host plants (chickpea, rice and wheat), which facilitated increased growth and yield [17,18,19]. Apart from IAA, gibberellic acid (GA), also known as diterpenoid acid, is a phytohormone that regulates seed germination, and the development of roots and shoots [20]. Similarly, the treatment with *Cladosporium* sp. of wheat and cucumber plants enhanced their growth due to induced GA production [21], and it has been suggested that GA production plays a crucial role in the host colonization of pea plants [22]. In addition, *Piriformospora* spp., *Phoma* spp., and *Trichoderma* spp. treatment elicited cytokinin (predominantly zeatin) production which promoted growth in *Arabidopsis* and melon [23,24]. It has also been well observed that certain PGPF (*Phoma* spp.) produce abscisic acid and increase plants’ growth, preferably under stress situations [8].

### 2.2. Plant Growth Promotion

Different groups of rhizosphere-resident fungi (*Trichoderma*, *Talaromyces*, *Rhizoctonia*, *Aspergillus*, *Penicillium*, *Fusarium*, etc.) tend to have a positive effect on augmenting the growth of various crop plants [8,10,25,26]. In contradiction to this, *Penicillium* spp. (*P. janthinellim*, *P. citreonigrum* and *P. citrinum*) residing in the native rhizosphere hampered the growth of native zinnia and tomato plants [8]. Shivanna et al. [10] reported that PGPF, such as sterile fungi, *Trichoderma* sp. and *Penicillium* sp., isolated from zoysiagrass specifically enhanced the growth of wheat plants of two different varieties. Thus, the reports suggested that rhizosphere-resident fungi exhibit unique specificity between host plants related to plant growth enhancement. Furthermore, non-sporulating fungi (SBF (sterile black fungus), SRF (sterile red fungus), and SDF (sterile dark fungus)) residing in the native rhizosphere soil of plants improved the growth and yield of corn, wheat, barley, chickpea, bromegrass, lupine, pea, clover, and ryegrass [26,27]. It has been noted that the time taken by PGPF is one of the crucial characteristics in improving plant growth and yield, and this statement was validated by the study of Muslim et al. [28], wherein PGPF (sterile fungi, *Trichoderma* sp. and *Penicillium* sp.) was reported to augment the growth of wheat plants in less than four weeks. However, in some plants, it has been stated that PGPF can take an even greater time to elicit plant growth.

PGPF treatments on plants have been well registered to enhance plant growth under greenhouse and field conditions [29,30,31,32,33,34]. Seed treatment with a conidial suspension of *Aspergillus niger*, *T. harzianum* and *Penicillium citrinum* has been reported to improve chickpea’s growth [17]. Furthermore, *Talaromyces* spp., *Chaetomium* spp., and *Exophiala* sp. maximized the growth of cucumber, chilli and brassica plants [22,23,24]. Furthermore, *Penicillium* sp. augmented chilli, tomato and sesame plants growth [35,36]. PGPF like *T. koningi* elicited not only growth but also the production of phytoalexin, and were successful in establishing root colonization in the case of *Lotus japonicus*. Apart from the above, the rate of photosynthetic activity increased in various crop plants upon PGPF treatments (*T. longipile*, *T. tomentosum*, *Aspergillus* sp., *Fusarium* sp., and *Penicillium* sp.) with increase in the total chlorophyll content, which is directly co-related to enhanced plant growth [30,35,36,37].

The root colonization ability of PGPF has resulted in increased root surface, lateral root numbers and root length, thereby enhancing plants growth [38,39,40,41,42]. In contrast to this statement, there are also reports stating that a few PGPF (sterile fungi) are not capable of colonizing plant roots to augment plant growth and improve crop yield [9,10]. If colonization is established, the fungus’ efficiency appears to be more in the upper part of the root region than the middle or lower regions [9,10]. Thus, the phenomenon of PGPF root colonization is suggested not to be a necessary criterion for the enhancement of plant growth specifically in the case of sterile fungi, and hence it has been stated to be a non-homogenous phenomenon. It may also be noted that the increased plant growth might be due to different chemical factors involved in the increase of plant growth treated with sterile fungi. The studies of Zavala-Gonzalez et al. [43] suggested that treatment with *A. ustus* promoted secondary root development by altering *A. thaliana* and potato plants’ root structure. It has been recently reported that treatment with PGPF (*Trichoderma* sp.) positively influences the enhancement of roots in pine plants [38]. In the studies conducted by Naziya et al. [7], seed treatment with PGPF caused increased in seed and vegetative growth parameters in chilli—apart from chlorophyll content—upon root colonization. Besides this, the PGPF treatment time for the induction of flowers plays a crucial role in the early fruit setting of agricultural and ornamental plants [44]. Treatment with *T. harzianum*, *P. chrysogenum*, and *P. simplicissimum* had an immediate action on early flower induction, and increased flower size and flower number in tomato and *Arabidopsis* plants [32,45]. The *Phoma* sp. treatment prominently increased plant height, leaf and fruit number in cucumber when tested under greenhouse conditions [28]. Besides this, *Rhizoctonia* sp., *Phoma* sp., *T. longibrachiatum*, *R. delemar* and non-sporulating fungus increased the yield of different crop plants [9,30,46,47,48,49].

Apart from the above, the elicitors extracted from the PGPF have been employed to elicit plants’ growth and resistance upon pathogen attack [50]. The elicitors include crude oligosaccharides, proteins, and sphingolipids, etc., which positively affect plant growth parameters. It has been noted that crude proteins extracted from PGPF improved the seed quality parameters predominantly in pearl millet and muskmelon plants compared to the control [6,29]. Similarly, volatile compounds produced by PGPF played a crucial role in promoting plant growth and development [29]. Treatment with terpenoids—such as the volatile compounds produced by *T. wortmannii* and *Phoma* sp.—elicited turnip and tomato plants’ growth, respectively [51,52]. The reports suggested that the volatile compounds produced by PGPF increased the lateral root development and the overall biomass of the plant, and induced early flowering in addition to maximizing the chlorophyll content [51,53].

Furthermore, different carrier materials have been examined in order to check the PGPF population and their effect on promoting plants’ growth. A study showed that treatment with *T. harzianum* talc-based bioformulation retains a good number of conidial suspensions compared to charcoal, sawdust, or cow dung, and vermiculite formulation with an extended shelf life and seed priming with the same prominently increased the root length, fresh weight, and overall biomass of chickpea and rice plants [54,55]. Besides this, a *T. harzianum* bioformulation prepared from the hydrolytic amino acids of pig corpses retained high PGPF conidial efficiency and significantly enhanced the chlorophyll content, shoot length, dry weight of shoots, and roots of cucumber, pepper and tomato plants. Different PGPF treatments and their effect on plant growth promotion have been listed in the Table 1.

### 2.3. Mineralization of Soil Nutrients

The mineralization and solubilization of complex organic phosphorous into their simple forms are known to be catalyzed by PGPF-produced enzymes, such as phytase, phosphatases, inorganic acids (HCl, nitric, sulphuric acids) and organic acids (α-keto-butyric, malic, glyoxylic, succinic, oxalic, fumaric, gluconic, tartaric, citric, 2-ketogluconic acids) [89,90,91,92,93,94]. The nutrient mineralization of soil mediated by PGPF plays a crucial role in promoting plant growth. It brings down the substrate’s degradation into a more soluble form for plants’ easy uptake [5,10,95]. Phosphate (P) is a vital nutrient that is well known to enhance plant growth. The fungi—namely *A. niger*, *A. tubingensis*, *P. bilaiae* and *P. oxalicum*—isolated from the rhizosphere soil, were successfully capable of solubilizing rock phosphates through the action of organic acid and phytase [63,80,96,97]. Besides this, *Penicillium* sp., isolated from the Indian Himalayan regions, also solubilized the complex phosphate [92]. Furthermore, *T. viride* has been reported to increase the organic soil carbon content in addition to high N, P, and K contents [70]. Treatment with *Penicillium* sp. and *Phoma* sp. enormously increased the absorption of N, P, and K in zoysiagrass and muskmelon plants [10,33]. Within PGPF, *Trichoderma* sp. has been exploited more to enhance nutrients and mineral absorption, chiefly Fe, N, P and K [8,10,98,99]. Different *T. harzianum* strains increased the accessibility of ammonium, nitrogen, zinc, copper, iron and manganese [5,100,101]. PGPF such as *Trichosporonbeigelii*, *C. albidus* var. *aurius*, *Phichia norvegensis* isolated from teff fields have been reported to solubilize phosphate, thereby positively enhancing the seed germination rate and vigour of faba bean [102].

According to a recent study, it has been stated that hydrogen cyanide (HCN) does not act as a biocontrol agent, but rather is associated with metal chelation, which in turn increases phosphate accessibility [103]. This PGPF competes for nutrients by decreasing Fe’s availability, and by limiting the growth of harmful microbes by producing low molecular weight compounds named siderophores, which create a race within soil microbes to acquire ferric ion, thereby acting as antagonists against harmful pathogens [11,104]. Plants take up iron through the chelation released by PGPF, and the released iron (siderophore-Fe complexes) is taken up instantly by plants through ligand exchange reactions [105]. Wolfgang et al. [106] reported a siderophore mixture of two PGPF (*P. chrysogenum* and *R. arrhizus*) upgraded Fe content, which prominently promoted growth in cucumber, maize and tomato plants, and increased the chlorophyll content with an increase in the supply of Fe EDTA. Machuca et al. [107] stated that pH and Fe concentration (III) play a crucial role in developing siderophores. The study proposed that the optimal pH range for siderophore production is between 6 to 8, and the ideal Fe (III) concentration ranges between 1.5 to 21 μM. It has been suggested that the increase in mungbean plant growth is due to the production of siderophore by PGPF—*A. parasiticus* and *A. niger*—at pH 7 [108]. Various PGPFs—namely *Absidia* spp., *Aspergillus* sp., *Thermoascus aurantiacu*s, *Aspergillus flavus*, *A. niger*, *A. tamarii*, *A. nidulans*, *Fusarium* sp., *Paecilomyces varioti*, *Cunninghamella* sp., *Penicillium spinolosum*, *P. indofıtico*, *P. oxalicum*, *P. chrysogenum*, *Rhizopus* sp., *Trichoderma* sp., *Beauveria* spp., and *Metarhizium*spp., etc.—have been reported to produce siderophores which have been reported to efficiently augment the growth of plants, and to effectively hinder the growth of pathogens [41,109,110,111,112].

Apart from the above, cyanide is also one of the crucial secondary metabolites produced during the early stationary stage of plant growth [113], in which amino acids (methionine, glutamate, glycine) serve as predecessors in the oxidative decarboxylation process [114]. *Aspergillus niger* and *Penicillium* spp., isolated from the native rhizosphere soils, are reported to produce HCN and ammonia, which positively impacts the enhancement of rice plant growth [115]. It has been reported that *Trichoderma* possesses the capability to hydrolyze ACC into ammonia, a major N (nitrogen) source for plant growth and development [116]. The ammonia and HCN produced by various *Trichoderma* isolates have documented plant growth promotion activity through the production of HCN [41,84,117].

### 2.4. Resistance against Stressors

#### 2.4.1. Antagonism

Antagonism is one mechanism wherein rhizospheric fungi tend to antagonize pathogens’ growth and development in plants. The antagonistic property of PGPF is suggested to be due to microbial predation, competition for nutrients, and antibiotic production, etc. [118]. The antagonism is suggested to be due to the production of lytic enzymes such as protease, chitinase and β- 1,3 glucanase [119]. Various PGPF—namely *T. harzianum*, *Phoma* sp., *F. equiseti* and *P. simplicissimum*—have been reported to be antagonistic against *R. solani*, *P. irregulare*, *S. rolfsii*, *F. oxysporum*, *P. syringae* and *C. orbiculare* [8]. Microbial predation mediated by PGPF involves the activation of the chitinase gene, which inhibits the growth of pathogens in an indirect manner [118]. PGPF such as *Gliocladium virens* produce an antibiotic called Gliovirin, which has restricted the growth of *Pythium ultimum* [120]. It has been suggested in the report of Kaur et al. [121] that *F. oxysporum* (non-pathogenic strains) isolated from rhizospheric soil inhibited the growth of pathogenic *Fusarium* strains via competition for nutrients, the reduction of chlamydospore germination, and competition to colonize infection sites (roots). Furthermore, *T. harzianum* suppressed the growth of *Pythium* sp., *R. solani* and *F. oxysporum,* thereby suggesting its antagonistic property towards many pathogens [5,122]. Similarly, *Aspergillus fischeri* displayed potent antifungal activity against *Botrytis cineria* [123]. To date, many fungi isolated from the rhizosphere have shown antagonistic responses towards pathogen growth [124,125,126]. However, reports state that antagonism need not be a specific criterion for the induction of resistance, as different rhizospheric fungi which are not antagonistic have also induced resistance in plants against various pathogenic infections [4]. This is because various other factors apart from antagonism may also play their part in the resistance to pathogen attack [4,7].

#### 2.4.2. Induction of Resistance

Induced resistance (IR) is generally demarcated as a phenomenon or a method of improving the plant’s inherent immune system. It is elicited upon treatment with an inducer or elicitor which is active against biotic stresses imposed by bacteria, fungi, viruses, parasites, and nematodes [127,128]. There are two forms of IR—namely induced systemic resistance (ISR) and systemic acquired resistance (SAR)—which are classified according to the basic nature of the inducer and the controlling defense signaling pathways [129]. ISR is an optimistic method to bring down the infection level in plants caused by deleterious pathogens, and the activation of ISR in plants upon pathogen infection is systemically expressed in the plant part that is spatially separated from the inducing area [130,131]. For example, the priming of plant roots with PGPFs can systemically induce resistance in leaf and shoots areas, and the activation of the same upon PGPF treatment is considered one of the safest and most cost-effective means of improving the crop plant’s growth and productivity [12,121,124].

Plant treatment with *Phoma* spp., *Penicillium* spp., *F. equisti*, *Trichoderma* spp. and other non-sporulating fungi induced systemic resistance by suppressing the growth of some soil-borne and airborne pathogens in different crops [5,40,128]. The treatment of different PGPFs and a non-pathogenic *F. oxysporum* strain (Fo47) induced systemic resistance to the invading pathogens in the case of cucumber plants [120,131]. Dong et al. [132] reported induced resistance against *Verticillium* wilt in upland cotton and sea-island cotton upon *P. chrysogenum* treatment. *P. simplicissimum* was reported to induce ISR responses in cucumber and *A. thaliana* plants [64]. PGPF (*A. ustus*) application triggered resistance against *B. cinerea* in *A. thaliana* and *S. tuberosum* [43]. *Trichoderma* sp. has been documented to induce ISR upon subsequent pathogen attack in different crop plants [13,133,134,135]. Likewise, treatment with *T. harzianum* reported induced systemic resistance against downy mildew pathogen in grapevine [136] and *T. asperellum* treatment improved resistance against *P. syringae* pv. *tomato* in the case of *Arabidopsis* [137]. Volatile compounds produced by three PGPF (*Cladosporium* sp., *Phoma* sp. and *Ampelomyces* sp.) induced subsequent systemic resistance in *Arabidopsis* plants upon infection with Pst (*P. syringae* pv. tomato) pathogen [51]. *Trichoderma harzianum*, *T. asperellum*, and *Talaromyces flavu*s induced resistance and successfully controlled sugar beet damping-off disease [138]. *Trichoderma,* along with other chemical treatments, showed successful control of head rot and root-knot diseases in the case of cabbage plants [139]. *T. asperellum* (T4 and T8) treatments showed a decrease in bacterial wilt disease incidence, along with an increase in tomato fruit yield. Besides this, the treatments also increased PAL, PPO, GLU enzyme activities through the activation of ISR upon infection with bacterial wilt pathogen in tomato plants [140]. Different PGPF—namely *Trichoderma* sp., *Aspergillus* spp., *Talaromyces* spp. and *Penicillium* spp.—showed the successful inhibition of anthracnose pathogen (*C. capsici*) in chilli. Furthermore, the tested PGPF—upon treatment—caused a reduction in anthracnose disease severity by ISR in chilli [7]. Thus, the PGPF-mediated protection in plants via ISR activation displays a high resistance towards pathogenic microbes in distant portions of the crop plants [77,99,141].

Systemic acquired resistance (SAR) refers to a distinct signal transduction pathway which includes the production of pathogenesis-related proteins and plays a crucial role in improving plant defense against pathogens [142,143]. In tobacco, the activation of SAR significantly reduced the occurrence of disease symptoms caused by various pathogenic microbes, namely *Phytophthora parasifica*, *Cercospora nicotianae*, and *Peronospora tabacina*, tobacco mosaic virus (TMV), tobacco necrosis virus (TNV), *P. syringae* pv. *tabaci* and *Erwinia carotovora* [144]. Treatment with *Trichoderma harzianum* induced PR protein production upon *Phytophthora capsici* infection in pepper plants [145]. A *Trichoderma* isolate (BHUF4) employed the SAR pathway and reduced the anthracnose infection caused by *C. truncatum* in chilli [146]; *Trichoderma asperelloides* enhances the SAR response under low nitrate nutrition in *Arabidopsis* [147]. Thus, PGPF-mediated SAR activation elicits a broad spectrum of systemic resistance in plants chiefly through PR protein production against disease-causing microbes [148,149].

The metabolites extracted from various fungi cause a great impression of imparting resistance in plants. The expression of elicitor-inducible PR proteins has been well correlated in imparting disease resistance [150]. Numerous glycoprotein elicitors have been extracted from *P. cinnamomic* [151], *Cladosporium fulvum* [152], *C. lindemuthianum* [153] and *R. solani* [154], which have been shown to induce resistance in plants. A protein elicitor (elicitin) (10 kDa) isolated from *Phytopthora* sp. was reported to trigger defense reactions in tobacco plants [155], along with a proteinaceous non-enzymatic elicitor from *T. virens* named Sm1, which efficiently elicited plant defense responses and induced systemic resistance against foliar pathogens [156,157,158]. Oligosaccharides extracted from PGPF are documented as strong elicitors in triggering defense reactions in crop plants against pathogenic microbes, and the underlying mechanism in the induction of resistance is reported to be reserved [159]. Cerebrosides, chiefly glycosphingolipids extracted from the *Fusarium oxysporum* f.sp. *lycopersici* (wilt-causing fungus) have been reported to elicit resistance against wilt disease in tomato plants. Cerebroside treatment was reported to reduce the anthracnose infection level in chilli plants in greenhouse conditions. The bio-preparation of fungal elicitors has been used efficiently to elicit plant resistance, including improving crop growth, crop yield and crop quality [29].

Bioformulations of *Trichoderma harzianum*, *T. asperellum*, *Talaromyces flavu*s successfully controlled sugar beet damping-off disease [138]. A combination of *Trichoderma* and a chitin-based bioformulation has been reported to successfully control the head rot and root-knot diseases of cabbage [139]. *T. asperellum* (T4 and T8) talc-based bioformulations showed a decrease in bacterial wilt disease incidence, along with an increase in tomato fruit yield. Besides this, bioformulations have been recorded to cause increases in PAL, PPO, GLU enzyme activities by ISR induction in the case of tomato against bacterial wilt [140]. The bioformulation of *T. harzianum* decreased the *Fusarium* wilt disease incidence in pumpkin plants [160]. Five PGPF, belonging to the genera of *Trichoderma* sp., *Aspergillus* spp., *Talaromyces* spp. and *Penicillium* spp. showed the successful inhibition of anthracnose pathogen (*C. capsici*) in chilli. Furthermore, the tested PGPFs have been documented to reduce anthracnose disease severity by ISR in chilli [4].

#### 2.4.3. Morphological and Histochemical Defense

The plant cell wall is the most crucial barrier for the pathogen to cross in order to gain access to the host plant. PGPF treatments strengthen the host plants’ cell wall by increasing the deposition of defensive wall materials, chiefly phenol, lignin, and callose [4,6]. One of the most vital challenges faced by plants is the attack of deleterious microbes; in response to this attack, plants trigger different mechanisms to confer resistance. The resistance mechanism includes the activation of a hypersensitive reaction (HR) and the biosynthesis of antibiotics, including phytoalexins that impart resistance in plants [161]. The HR is the localized necrosis (cell death) of plant cells, which chiefly occurs at the infection site and is supposed to limit the spread of invading pathogens [162]. It occurs during resistance interactions between pathogens and host plants; during the process, brown necrotic spots can be seen at the site of the pathogen infection [132]. The primary event of HR includes the rapid production of ROS compounds, leading to the production of ROS like hydroxyl radicals, superoxide, and hydrogen peroxide (H_2_O_2_) [163].

The deposition of H_2_O_2_ in cell membranes is caused by the production of ROS at the site of infection, thereby inhibiting the growth and development of the pathogen [164]. The ROS production is toxic, and it has a lethal effect on microorganisms and the plant’s own cells (in higher concentrations). Still, this effect is further effectively induced by various antioxidant enzymes, of which the activity is elevated upon PGPF treatment under biotic and abiotic stress [14]. Different reports have been documented, stating that callose deposition occurs during resistance interactions upon pathogen infection, and it can also occur during pre-treatment with some chemical or biotic inducers, which elicits the innate protective resistance mechanism in plants [165,166]. Yedidia et al. [167] reported that *T. harzianum* T-203 enhanced callose deposition in cucumber seedlings. Similarly, callose deposition in the roots of *A. thaliana* seedlings was promoted by *T. harzianum* [168]. Pearl millet seedlings treated with *T. hamatum* showed high lignification and callose deposition upon infection with a downy mildew pathogen [169]. Furthermore, the *T. atroviride* TRS25 strain maximized the callose deposition in cucumber plants upon *Rhizoctonia solani* infection [15], and the treatment with *Talaromyces funiculosus* increased the deposition of callose in the epidermal walls of chilli seedlings upon *C. capsici* infection [4].

Lignin is a secondary metabolite produced in plant cells by the phenylalanine/tyrosine metabolic pathway, which acts as a potential barrier against deleterious pathogens. It triggers defense in plants upon pathogen entry by depositing lignin across the entire cell wall or on a set of cells, or only at the site of pathogen infection [170]. Lignification was induced in cucumber seedling hypocotyls with different PGPF culture filtrates [131]. Cucumber plants treated with *Penicillium simplicissimum* GP17-2 showed enhanced lignin formation [171]. Triple combinations of *Pseudomonas*-PHU094, *Trichoderma*-THU0816 and *Rhizobium*-RL091 showed an increase in the deposition of lignin in the case of chickpea plants which were previously infected with *S. rolfsii*. The deposition was high in the interfascicular cells of cambium, particularly in the sclerenchyma cells of phloem. Thus, PGPF, when employed individually, enhanced physical strength and cell wall durability against phytopathogens, but the effect can be elevated when it is applied in a synergistic consortium [54,172,173].

#### 2.4.4. Biochemical Defense

The plant’s innate immunity to pathogen attack is also associated with defense-related biochemical mechanisms other than morphological and histological modifications, as various biochemical changes occur during the process. Interactions between the pathogen and the host plant encourage changes in cell metabolism and enzyme activities, such as phenylalanine ammonia-lyase (PAL), peroxidase (POX), lipoxygenase (LOX), superoxide dismutase (SOD), β-1,3 glucanase, and polyphenol oxidase (PPO), etc. [127,174]. Seed treatment with specific PGPF types, such as *Rhizoctonia* spp., *Fusarium* sp., *Aspergillus* spp., *Phoma* spp., *Talaromyces* spp., *Trichoderma* spp., *Pythium* spp., *Penicillium* spp., and non-sporulating fungus increased the rates of POX occurrence in cucumber, pearl millet, and chilli [4,98,172,175]. Inoculation with *Trichoderma* spp. increased the peroxidase and chitinase activities in the roots and leaves of cucumber plants [167]. The high PPO level is evidenced by beneficial fungi in banana when their roots were treated with *F. oxysporum* [176]. *Phoma* spp. and non-sporulating [GU21-2] fungus have been shown to maximize the PPO activity in cucumber plants upon treatment [172]. Similarly, the *Trichoderma* strain elevated PPO enzyme activity in chilli, pigeon pea, and moong bean upon challenge inoculation with *C. capsici*, *F. oxysporum* and *Alternaria alternata* [175,177].

PGPF, like *Trichoderma* sp., elevated the expression of LOX activity in the case of pearl millet [169], and root colonization by *Trichoderma* isolates has been reported to elevate the levels of peroxidases, chitinases, and β-1-3-glucanases [167]. A significant elevation in the mRNA levels of Chit1, the β-1,3-glucanase gene and the peroxidase gene was observed in the leaves of *Trichoderma*-induced plants at 48 h post-challenge inoculation with *P. syringae* pv. *lachrymans* [12]. A rise in chitinase activity was instigated in the presence of PGPF, i.e., *U. atrum*, which showed biocontrol properties [178]. *T. harzianum*Tr6 isolated from the cucumber rhizosphere were examined as both single and combination treatments, which induced resistance in cucumber plants against *F. oxysporum* f. sp. *radices cucumerinum* infection. This induced resistance is reported to be accompanied by a primed expression of the defense-related genes (CHIT1, β-1,3-Glucanase, PAL1 and LOX1) upon challenge inoculation with *Fusarium* [179]. *Phoma* spp. [GS8-1] and nonsporulating [GU21-2] fungus increased the chitinase activity in cucumber upon inoculation with *Colletotrichum orbiculare* [180]. *Trichoderma* elicited a high level of β-1,3 glucanase and chitinase enzyme activities, which induced resistance in *Capsicum* and French bean plants [41]. PGPF (*Phoma* sp.) and non-sporulating fungus GU21-2 treatment caused high glucanase activity in cucumber [172]. Seeds primed with *Penicillium* sp. have induced increased chitinase activity in the case of pearl millet upon pathogen inoculation [99]. *Trichoderma atroviride* treatment on cucumber plants increased PPO enzyme activity upon challenge inoculation with *Rhizoctonia solani* [13]. The application of crude metabolites isolated from various fungi stimulated plant defense responses by the induction of defense-related enzymes, chiefly PAL, POX, PPO and LOX [181]. Cycloproteins were reported to induce systemic resistance in *Nicotiana benthamiana,* and reduced *P. nicotianae* and TMV infection [182].

#### 2.4.5. Defense Signaling

During local and systemic defense responses, a large set of defense enzymes, PR proteins, and signal molecules are manufactured in plants in order to elicit a wide range of antimicrobial activity [183]. The transcript accumulation pattern of defense enzymes during PGPF-mediated induced resistance is explored less and has received less attention. Defense responses triggered by beneficial and parasitic microorganisms regulate defense signaling networks, wherein the plant hormones SA, JA, and ET play the chief role [184]. There is ample evidence that displays the cross-communication between SA, JA and ET pathways to fine-tune the plant defense responses based on the invader encountered [185]. Cucumber plants pretreated with *T. asperellum* T203 were reported to activate JA/ET-systemic resistance linked with potential PR gene expression in response to pathogen infection [12].

A few reports have stated the involvement of SA-dependent signaling upon *Trichoderma* sp. treatment [186,187]. Tomato plants pretreated with *Trichoderma* were reported to elevate the expression of JA-responsive genes, which triggered systemic resistance against *B. cinerea* [12]. Different studies have suggested that ISR activation by *Trichoderma* sp. is involved in JA and ET signaling [11,188]. Moreover, the activation of both SA and JA pathways has also been documented for a few *Trichoderma* strains [43,137]. Many *Trichoderma* sp. have been stated to break down the cellulosic plant biomass with the cellulase enzyme’s action [189], which instigate ISR responses in plants such as tobacco, lime, bean, corn and cucumber by eliciting the ET or JA defense pathway [190,191,192]. Similar effects have been observed upon treatment with protein LSP1 and polysaccharide extracted from *M. guilliermondii*, wherein the treatments stimulated mitogen-activated protein kinases, defense-related gene expression, and defense signaling accompanied by glycyrrhizic acid biosynthesis [193]. Different effects of PGPF treatments in elicitation of resistance towards deleterious pathogens have been represented in the Table 2.

#### 2.4.6. PGPF in Abiotic Stress Improvement

Crop plants encounter different abiotic stress challenges—including excessive temperature, drought, salinity, floods, and heavy metal accumulation—which exclusively affect crop plant growth and yield [201,202]. Most cultivated lands face one or more of the above-listed stresses, which decrease crop plants’ yields by up to 70% [203]. Climate change threatens the future loss of crop plant productivity, predominantly cereal crop plants, greatly impacting food security [204]. Most agricultural lands showed a high saline content (up to 37%) from 1990 to 2013 [205]. PGPF-mediated growth promotion in plants under stress or pressure conditions is reported to be due to the root construction changes, mineral solubilization from the dead organic substances, and (secondary) metabolite production [10]. PGPF that is naturally harbored in the soil system possesses the capacity to strengthen the plant’s immune system and improve plant growth under stressful conditions [5]. *T. harzianum-*treated tomato plants offered early seed germination and seedling vigor with an increased shoot and root length, and shoot fresh weight under biotic, abiotic and physiological pressures [206]. *Trichoderma asperellum* (siderophore producing strain) promoted cucumber plants’ growth under salt stress [207]. *Penicillium* sp., isolated from the rhizosphere soil of peanut, improved the saline tolerance ability in sesame plants [208]. *Trichoderma* spp. elicited abiotic stress tolerance ability against biotic pressure imposed by phytopathogens [209]. *T. atroviride* treatment improved the drought tolerance ability of maize plants with increased antioxidant enzyme machinery [210]. *T. hamatum* treatment triggered growth and the drought tolerant capacity in *Theobroma cacao*. *T. harzianum* treatment mitigated the salt (NaCl) tolerance capacity in Indian mustard plants with maximized antioxidant enzyme defense machinery [211].

Different groups of PGPF—such as *Microsphaeropsis*, *Mucor*, *Steganosporium*, *Phoma*, *Aspergillus*, *Alternaria* and *Peyronellaea* were—have been reported to protect *Arabidopsis* plants from heavy metal accumulation [212]. *T. harzianum* improved salt, osmotic, heat and oxidative stress conditions in *A. thaliana* plants with an increase in heat shock protein, and APX and SOS transcript activities [213]. *T. virens* improved the cadmium tolerance ability in *N. tabacum* plants with a reduction in lipid peroxidation and increased antioxidant enzyme activity [214]. *T. harzianum* treatment increased the drought tolerance ability in *N. tabacum* plants with an increase in their relative water content, along with a decline in their transpiration rate [215]. Furthermore, it elicits antibiotic production, mycoparasitism, opposition, and ISR activation [216]. PGPF strengthens the cell wall of plants and prevents solute loss during abiotic pressure [217]. During stress conditions, callose accumulation increases the plugging of the sieve pores and improves plasma membrane deposition and cell wall apposition [218]. Lignin is involved in the plant defense against varied sets of causative agents, including pests, and its effect is elevated in plants when inducing tolerance to different stressful situations (heavy metals, salinity, high or low temperature, drought and other pressures) [219]. Saline tolerance by PGPF is brought about by improving the sterol content for fatty acid enzyme modification [220,221]. PGPF are both salt-loving and salt-tolerant due to their inherent osmotic conditions [222]. PGPF interaction was affected by atmospheric and host factors, wherein the fungal population was found to be associated with the salinity rates [223]. Different fungi have been isolated in varied saline conditions, indicating their survival potential [224]. A sodium chloride concentration of ≥20% is tolerated by *Penicillium* and *Aspergillus* spp., and this PGPF resilience to adverse environment pressures is due to the production of osmotic substances [225].

Heat tolerance ability has been imparted by *C. protuberata* in *Dichanthelium lanuginosum*, but both the fungus and the plant showed no survival ability above 38 °C [226]. Different strains of *Penicillium* sp. were reported to boost plant immunity in sesame plants upon *Fusarium* infection under salt stress conditions [208]. Heavy toxic metals bring out cytoplasmic enzyme inactivation and cause injury to the cell membrane, resulting in low plant growth [227]. *Penicillium resedanum* induced GA production, which promoted chilli plants’ growth under abiotic pressures [35]. *Trichoderma* sp. has been reported to possess a tolerance ability toward heavy metals, chiefly zinc, copper, lead, cadmium, and so on [228,229]. The instant build-up in the betaine, glycerol, and proline content chiefly in intracellular spaces has shown resistance to high salinity [222]. PGPF also elicits tolerance to varied abiotic pressures due to increased proline content and other enzyme (PPO, catalase, SOD, APX, etc.) activities [42,211]. A common mechanism through which PGPF improved the level of tolerance under abiotic pressures could be the amelioration of destruction triggered by ROS (reactive oxygen species) accrual in the case of stressed plants [206]. *T. harzianum* elicited ROS production via ISR activation by Thph 1 and 2 proteins in the case of maize [191]. Therefore, the ROS effect can be combated upon PGPF treatment, which has ROS scavenging capability to recycle ascorbate and glutathione products under abiotic pressures by the activation of antioxidant defense enzyme machinery [210,211]. Proline content is reported to efficiently cause an increase to the induced tolerance to various pressures in plants. Different enzymatic and non-enzymatic antioxidant machineries have been reported to enhance PGPF treatment, strengthening the phytoconstituent composition and protecting the plants from the extra damage under abiotic pressures [211,230]. PGPF treatment was reported to limit the ethylene (ET) concentration by increasing ACC deaminase activity, which transformed ET into NH_4_ and α-keto- butyrate upon root colonization in plants [231]. Different reports have stated that PGPF colonization elevated ACC deaminase levels and augmented crop plants’ growth under stress situations. For example, *Trichoderma* spp. produced ACC-deaminase, which regulates the endogenous ACC level, thereby eliciting root elongation and improving plants’ inherent resistance capacity against abiotic pressures [42]. According to a recent report, *T. longibrachiatum* (T6) enhanced wheat plant growth and increased the capacity level to tolerate NaCl pressure by increasing ACC-deaminase enzyme activity (lowering ethylene) and IAA gene expression. This alleviated the effect caused Na^+^ damage and enhanced Na^+^ or H^+^ (antiporter gene) in wheat plants [232]. The tolerance effect of PGPF against various abiotic stresses in different plants has been listed in the Table 3.

### 2.5. PGPF as a Source of Alternatives

The disease is not the only outcome of plant-microbe interactions. Plants do have defense mechanisms that they utilize themselves, both locally and systemically. Several mutually beneficial relationships between plants and microbes affect agricultural productivity and plants’ health in general. These systems have also been the foci of intensive studies. In symbiotic relationships, the microbe assists the plant with nutrient absorption, or contributes biochemical and molecular activities that the plant lacks [98]. The plant, in turn, contributes photosynthate, to the competitive advantage of the corresponding microbial symbiont in the rhizosphere. By altering the balance of microflora in the rhizosphere, symbiotic associations may also help protect plants from disease-causing microbes [7]. The exploitation of other beneficial, non- symbiotic rhizosphere organisms for the biological control of plant diseases is also an important discipline which relies upon detailed knowledge of specific plant–microbe interactions and the general ecology of interacting microbes in the soil.

Presently, most of the research studies are focused on combating plant diseases through chemical insecticides or pesticides and fungicides, which are lethal to beneficial soil microflora and are also not safe for human consumption. Moreover, many chemical fertilizers, chiefly N, P, and K, have been applied to improve various crop plants’ growth, which causes soil pollution [240]. Therefore, different eco-friendly strategies have been put forth by research communities in the last decade to minimize the use of chemical-based fertilizers in crop improvement and management studies. As biocontrol agents, endophytes, Arbuscular mycorrhizae (AM) fungi, plant extracts, and PGPR have been explored widely as substitutes for controlling plant diseases caused by deleterious microbes [3,4,34,241,242]. PGPF is the least-explored emerging eco-friendly source for combating different bacterial and fungal diseases in plants, in place of other biocontrol agents. PGPF is highly significant in enhancing plant growth, fruit yield and nutrient composition, and boosting plants’ immunity by increasing their defense machinery, which regulates pathogen infections [4,5,6,7,34,44]. This efficiency of PGPF can be attributed to its efficient radical colonization capability, mineral solubilization (PGP traits), activation of antioxidant and defense-related enzymes, and production of antibiotics, phytohormones and volatile compounds under various biotic and abiotic stresses. However, the selection of the potential PGPF is vital, as the plant responses to environmental conditions vary based on different aspects, e.g., plant genotype, the site of evaluation, and seasons, etc. To date, the exploitation of PGPFs has been regarded as a promising substitute for the present-day practices, and it needs to be further exploited for sustainable agriculture.

## 3. Conclusions

PGPF applications are highly efficient in enhancing crop yield, crop quality, and the sustainable management of plant diseases. The treatment with PGPF will reduce the cost and pollution effect caused by chemical fertilizers. PGPF can be used as a substitute to reduce the use of agrochemicals in plant disease management. The use of PGPF improves plant performance and crop yield in agricultural fields. Information about PGPF and host plant interactions with the pathogen infection helps us understand the signaling mechanism involved in plant growth and resistance. The induction of resistance mediated by PGPF is linked with the production of phytohormones, defense-related enzymes, antibiotics and other signaling hormones, which play a crucial role in offering long-time protection and resistance during biotic and abiotic pressures. Thus, the treatment of PGPF increased crop yield production and reduced the disease-causing effect of pathogens in agricultural crop fields.

## Figures and Tables

**Figure 1 jof-07-00314-f001:**
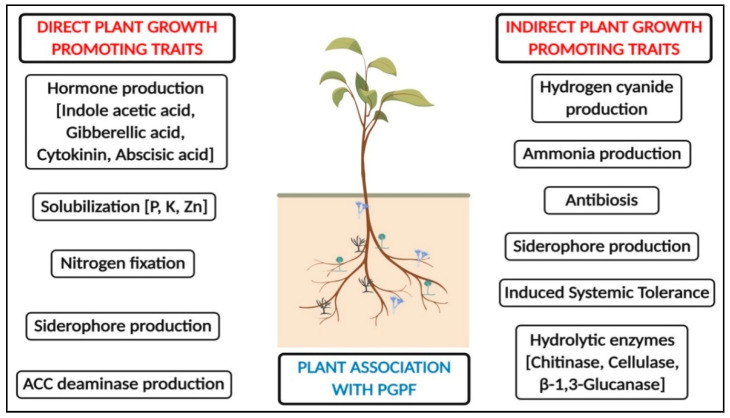
Outlook on the beneficial characteristics of rhizosphere-resident plant growth promoting fungi (PGPF) for plant growth.

**Figure 2 jof-07-00314-f002:**
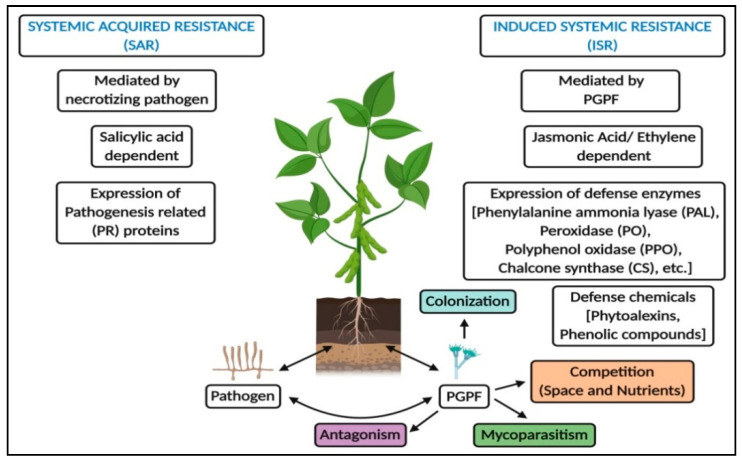
Mechanism of resistance offered by PGPF against the invading pathogens.

**Figure 3 jof-07-00314-f003:**
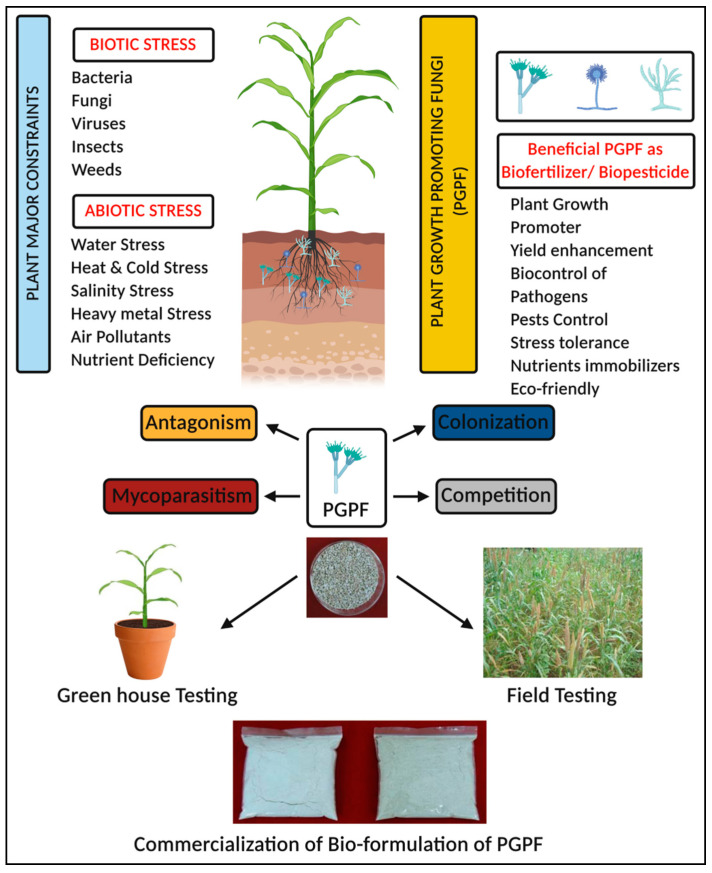
Application of PGPF and its effect on the induced resistance in plants.

**Table 1 jof-07-00314-t001:** Mechanism of plant growth promotion mediated by PGPF in different crop plants.

PGPF	Plant	Effect	Reference
*R. nigricans*, *F. roseum*	*Solanum lycopersicum*	Enhanced shoot dry weight	[56]
*T. harzianum*, *T. koningii*	*Solanum lycopersicum* and *Nicotiana tabacum*	Enhanced dry weight and improved seed germination	[57]
*T. ciride*	*Solanum lycopersicum*	Maximized plant height	
*T. harzianum*	*Vinca minor*	Improved flowering, weight and height of the plant	[58]
Sterile dark fungus (SDF)	*T. aestivum*	Increased shoot dry weight	[59]
*Penicillium simplicissimum* (GP17-2)	*Cucumis sativus*	Increased root and shoot growth	[4]
*T. harzianum* (T22)	*Zea mays*	Increased shoot growth, root area and root size	[60]
*A. niger*	*B. chinensis*	Increased plant dry weight and N, P content	[61]
*Penicillium* sp.	*Triticum aestivum*	Solubilized phosphate	[62]
*Penicillium janthinellum* and *Penicillium simplicissimum*	*Arabidopsis thaliana*	Elevated shoot biomass and leaf number	[63]
*Fusarium equiseti* (GF19-1)	*Lycopersicon lycopersicum*	Increased plant biomass with more root and shoot growth	[64]
*Penicillium citrinum*	*Suaeda japonica*	Increased root and shoot length	[65]
*Trichoderma virens*	*Arabidopsis thaliana*	Improved biomass and lateral roots development with the production of IAA	[66]
*Trichoderma harzianum* (GT3-2)	*Cucumis sativus*	Increased root and shoot growth	[67]
*Phoma herbarum* and *Aspergillus fumigatus*	*Glycine max*	Increased plant height and plant biomass. Maximized shoot growth, leaf area and chlorophyll content	[68]
*Trichoderma viride*	*Saccharum officinarum*	Improved crop yield	[69]
*Fusarium equiseti*	*Spinacia oleracea*	Improved overall plant biomass by maximizing root and shoot growth	[70]
*Trichoderma viride*	*Gossypium arboreum*	Increased root, shoot length and plant dry weight	[71]
*Fusarium equiseti* (GF19-1)	*Cucumis sativus*	Increased root and shoot growth	[72]
*Trichoderma harzianum*	*Cucumis melo*	Induced early seed germination and increased seedling vigor	[73]
*Fusarium oxysporum*(MSA-35)	*Lactuca sativus*	Increased root and shoot growth with high chlorophyll content	[74]
*Penicillium simplicissimum*	*A. thaliana* and *N. tabacum*	Maximized shoot fresh weight, shoot dry weight and leaf number	[75]
*Trichoderma* spp.	*Lycopersicon lycopersicum*	Increased dry matter biomass and improved overall plant growth	[76]
*Trichoderma harzianum*(T-22)	*Prunus cerasus* × *P. canescens*	Increased root growth and development	[77]
*Sphaerodes mycoparasitica*	*Triticum aestivum*	Increased seed germination and seedling vigour	[78]
*T. harzianum* (T-75)	*Cicer arietinum*	Increased crop yield	[79]
*Alternaria* sp., *Phomopsis* sp., *Cladosporium* sp., *Colletotrichum* sp., *Phoma* sp., *Aspergillus* sp. and	*Nicotiana tabacum*	Improved overall plant growth biomass by maximizing the root and shoot growth and high chlorophyll, soluble sugar content	[80]
*Trichoderma* sp.	*Phaseolus vulgaris*	Positive for plant growth promoting traits, i.e., phosphate, siderophore, HCN and Ammonia	[81]
*Trichoderma longibrachiatum*	*Triticum aestivum*	Increased plant height, root length, shoot fresh and dry weights. Increased chlorophyll a, b and total chlorophyll content.	[82]
*Trichoderma* sp.	*Lycopersicon esculentum*	Produced IAA, siderophore, HCN, ammonia and solubilized phosphate	[83]
*Penicillium* spp. GP16-2	*N. tabacum*	Enhanced shoot fresh, dry weight and increased the leaf number	[84]
*Aspergillus terreus* JF27	*Lycopersicon esculentum*	Enhanced fresh weight and shoot length	[85]
*Alternaria* sp.	*Salvia miltiorrhiza*	Enhanced fresh weight and dry weight	[86]
*T. harzianum*	*Lycopersicon esculentum*	Increased root and shoot growth. Maximized leaf area and vigour of tomato seedlings. Elicited the production of IAA and GA.	[87]
*T. viride* and *T. harzianum*	*Citrullus lanatus*	Increased the number of leaves, leaf dry weight, stem length and the number of branches. Enhanced chlorophyll content and N, P, K uptake. Also enhanced the fruit number, seeds number, fruit weight and dry weight.	[49]
*Phoma* sp.GS 8-3, *Trichoderma asperellum* SKT-1, *Fusarium equiseti* GF18-3 and *Penicillium simplicissmum* GP17-2	*Allium cepa*	Enhanced the plant height, root length, bulb perimeter and plant dry weight.	[88]

**Table 2 jof-07-00314-t002:** Mechanism of resistance mediated by PGPF in different crop plants.

PGPF	Plant	Effect	Reference
*C. cucumerinum* and *F. roseum*	*S. tuberosum*	Increased lignin deposition	[194]
*Trichoderma harzianum* (T-203)	*Cucumis sativus*	Enhanced callose deposition	[153]
*F. oxysporum*	*Lycopersicon esculentum*	Induced resistance by reducing the incidence of *Fusarium* wilt disease	[195]
*F. oxysporum*	*Asparagus officinalis* *Lycopersicon esculentum*	Exhibited HR response, increased defense related enzymes activity PO, PAL, lignin content and reduced the disease severity upon pathogen infection	[196]
*Pectobacterium atrosepticum* and *P. infestans*	*Solanum tuberosum*	Elicitors obtained from the culture filtrates induced high levels of phenolic compound and PAL enzyme activity	[197]
*Trichoderma* sp.,	*Cucumis sativus*, *Arabidopsis thaliana*.	Triggered SA and JA/ET pathways in eliciting defense in plants	[198]
*Penicillium* spp.	*P. glaucaum*	Enhanced protection against downy mildew pathogen	[25]
*Trichoderma harzianum*	*Cucumis sativus* and *Arabidopsis thaliana*	Induced systemic resistance in cucumber and *A. thaliana* with defense related genes expression	[164]
*Phoma* sp., *Penicillium* sp., *Trichoderma* sp., *Aspergillus* sp., *Fusarium* sp.	*Lycopersicon esculentum*	Enhanced protection by reduction of disease against *R. solanacearum* with increasing the activity of defense related enzymes PAL, POX and GLU activities	[32]
*Penicillium simplicissimum*	*Cucumis sativus*	Induced protection against anthracnose disease showing a reduction in the lesion number and lesion diameter. Elevated the activity of exo-glucanase, exo-chitinase, PO and PPO	[171]
*Phoma* spp.	*Cucumis sativus*	Induced resistance by increasing the chitinase activity upon infection with *Colletotrichum orbiculare*	[180]
*T. harzianum*	*Arabidopsis thaliana*	Elicited callose deposition in the roots of seedlings	[168]
*Trichoderma viride*	*Cajanus cajan*, *Vigna radiata* and *Vigna mungo*	Elevated the levels of ROS and increased the levels of PO, PPO and PAL activities. Maximized SOD, CAT, AOX activities and total phenolics. Reduce the severity of diseases against *Fusarium oxysporum* and *Alternaria alternata*	[177]
*Phoma* sp. and non-sporulating fungus	*Cucumis sativus*	Induced protection against anthracnose disease under pot and field conditions. Inhibited the germination of *C. orbiculare* pathogen. Elevated the formation of lignin and expression of PAL, PO, PPO, GLU and CHI activities upon pathogen infection	[172]
*Penicillium* spp. GP16-2	*N. tabacum*	Decreased disease severity of Cucumber Mosaic Virus (CMV) in tobacco plants	[75]
*Trichoderma* spp., *T. viride*, *T. harzianum*	*Lycopersicon esculentum*	Antagonize the growth of *A. solani* (early blight pathogen) with a reduction in the percent disease index	[199]
*Rhizoctonia* sp.	*Cucumis sativus*	Induced systemic resistance against *C. orbiculare* by reducing the total lesion number and total lesion diameter along with increasing the activity of POX enzyme	[173]
*A. niger*	*Glycine max*	Triggered callose deposition in leaves and roots upon infection with *Heterodera glycines*	[200]

**Table 3 jof-07-00314-t003:** Mechanism of abiotic stress tolerance mediated by PGPF in different crop plants.

PGPF	Plant	Effect	Reference
*Trichoderma harzianum* (1295-22)	*Zea mays*	Mitigated oxidative stress caused by sodium hypochlorite and lipid peroxidation in sweet corn	[233]
*Trichoderma harzianum* (T-22)	*Lycopersicon lycopersicum*	Improved seed germination under stress	[35]
*Exophiala* sp. LHL08	*Cucumis sativus*	Improved growth under salinity and drought stresses	[234]
*Trichoderma harzianum*	*Lycopersicon esculentum*	Increased plant growth biomass and Induced resistance to water deficit. Enhanced APX, CAT and SOD activity. Improved the ability of plants damage caused by ROS	[235]
*Trichoderma harrzianum* and *Fusarium pallidoroseum*	*Oryza sativa*	Higher biomass production and increased induction of SOD, CAT and POD	[236]
*Trichoderma hamatum*	*Ochradenus baccatus*	Alleviated abiotic salt stress by improving plant growth and antioxidant defense enzyme activity	[237]
*Penicillium* sp.	*Sesamum indicum*	Increased root and shoot length, maximized fresh and dry weight of seedlings under salt stress. Increased amino acid, chlorophyll a,b and total chlorophyll content. Also enhanced protein and nitrogen content. Induced protection against *Fusarium* wilt disease	[206]
*Trichoderma harzianum*	*Brassica juncea*	Mitigated NaCl stress by enhancement of antioxidant defense machinery. Improved shoot, root length and plant dry weight	[211]
*Trichoderma longibrachiatum*	*Triticum aestivum*	Increased the tolerance of plants to salt stress by SOD, POD, CAT gene expression	[82]
*Trichoderma atroviride*	*Zea mays*	Ameliorated drought stress by enhancement of antioxidant defense in plant seedlings	[210]
*Trichoderma harzianum*	*Cucumis sativus*	Improved defense by alleviated oxidative and nitro-stative stress by minimizing ROS production and RNO species production upon infection with *F. oxysporum* by enhancing the antioxidant potential	[238]
*Talaromyces* sp., *Penicillium* sp., *Mucor* sp., *Fusarium* sp., *Pestalotiopsis* sp., *Aspergillus* sp., etc.	*Oryza sativa*	Improved the growth of plants and antioxidant capability, also, to increase in proline and soluble sugar content	[239]

## Data Availability

The data presented in this study are available in this manuscript.

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
