# Peer review of "Bioprospecting of Rhizosphere-Resident Fungi: Their Role and Importance in Sustainable Agriculture"

_jof, 2021, doi:10.3390/jof7040314_

Round 1
Reviewer 1 Report
The theme discussed by the authors is undoubtedly very important since the application of PGPF will reduce the use of chemical control to minimal and protects plants against various biotic and abiotic stress. In summary, this review presents a holistic view of PGPF as efficient natural biofertilizers for the improvement of growth and resistance in crop plants. The overall interest of the work is certainly very high.
Likewise, the presentation and the arrangement of the manuscript are quite correct. The language level is good and does not need any serious corrections. Authors collected a large volume of information concerning the PGPF. Under this approach, the submitted review-article presents interest.
At the same time, the review in the current form has some drawbacks that should be addressed prior it can be accepted for publication. The most serious of them are the following:
First. In my opinion, it would be very useful to include an initial table with the explanation of all the acronyms use in the review.
Second. The main value in any review is the analytical part. Authors described a lot of information and compiled the main findings into tables, and this is good way for information presentation. However, in my opinion, the authors should include a brief discussion at the end of each section. They could discuss the collected information.
Third. Some relevant papers are not mentioned in the long catalog of references and should be added:
Sánchez-Montesinos, B.; Diánez, F.; Moreno-Gavira, A.; Gea, F.J.; Santos, M. Plant growth promotion and biocontrol of Pythium ultimum by saline tolerant Trichoderma isolates under salinity stress. Int. J. Environ. Res. Public Health 2019, 16, 2053.
Sánchez-Montesinos, B.; Diánez, F.; Moreno-Gavíra, A.; Gea, F.J.; Santos, M. Role of Trichoderma aggressivum f. europaeum as plant-growth promoter in horticulture. Agronomy 2020, 10, 1004.
Fourth. Authors must review the format of the bibliographic references and adapt them to the format required by the journal.

Author Response
RESPONSE TO “REVIEWER 1” COMMENTS
We profusely thank the reviewers for their constructive comments. Herewith we are submitting the revised manuscript following incorporation of all the suggestions as indicated by the reviewers. We hope the reviewers will be happy with the corrections incorporated. In the revised manuscript, the questions raised have been addressed and the changes are made in red colour throughout the manuscript.
Comment No. 1. First. In my opinion, it would be very useful to include an initial table with the explanation of all the acronyms use in the review.
Answer: As per the suggestion, a list has been added initially with explanations of all the acronyms in the revised manuscript.
Comment No. 2. Second. The main value in any review is the analytical part. Authors described a lot of information and compiled the main findings into tables, and this is good way for information presentation. However, in my opinion, the authors should include a brief discussion at the end of each section. They could discuss the collected information.
Answer: As per the suggestion, a paragraph has been added in the end of the manuscript discussing all the roles of fungi in improving the plant health and productivity in the section “PGPF as a source of alternatives”.
Comment No. 3. Third. Some relevant papers are not mentioned in the long catalog of references and should be added:
Sanchez-Montesinos, B.; Dianez, F.; Moreno-Gavira, A.; Gea, F.J.; Santos, M. Plant growth promotion and biocontrol of Pythium ultimum by saline tolerant Trichoderma isolates under salinity stress. Int. J. Environ. Res. Public Health 2019, 16, 2053.
Sanchez-Montesinos, B.; Dianez, F.; Moreno-Gavira, A.; Gea, F.J.; Santos, M. Role of Trichoderma aggressivum f. europaeum as plant-growth promoter in horticulture. Agronomy 2020, 10, 1004.
Answer: As per the suggestion, the quoted references have been added in the revised manuscript.
Comment No. 4. Fourth. Authors must review the format of the bibliographic references and adapt them to the format required by the journal.
Answer: As per the suggestion, the bibliographic references have been formatted according to the journal format.

Reviewer 2 Report
The manuscript entitled „Bioprospecting of Rhizosphere Resident Fungi: Role and Importance in Sustainable Agriculture” is a work reviewing different aspects of biological control. It includes a large amount of data. However, in my opinion, the work should be improved before considering for publication in another journal. A lot of information included in the manuscript is well known and described and summarized previously several times. Moreover, in the manuscript, no further research trends, are suggested. The authors do not ask any questions that have not been answered yet. Additionally, often the content of subsections does not fit the subsection titles.
Therefore, in my opinion, the article is not suitable for publication in this form.
Below, I present some additional suggestions and comments.
The subsection „Antagonism” is unfortunately chaotically written. It presents some examples of different plant-microorganism interaction, but much of them is not an example of antagonism.
In the subsection, describing induction of resistance, authors write about Systemic Acquired Resistance (SAR), but they do not present any examples of this type of resistance.
In the subsection „Morphological and Histochemical Defense”, the authors also describe the biochemical response of plants to pathogens, including ROS and phenolic compounds generation. The name of the subtitle should be changed or some information should be transferred to the next subsection.
In my opinion, the section „PGPF as a source of alternatives” which, taking into account the title of the manuscript, should be the longest part, is too short.
Authors should use full names or abbreviations consequently throughout the text. Moreover, they should avoid colloquial phrases.
Line 71: Is „signalling”, should be „signaling”
Line 101: Is „cytokinine”, should be „cytokinin”
Line 120: Is „to augment”, should be „augmenting”
Line 132: Is „to increase”, should be „to the increase”
Line 149: Is „to increase”, should be „increasing”
Line 184 – 192 should be not included in the manuscript
Line 218: What is HCN?
Line 218: Is „that,”, should be „that”
Line 219: Is „increase”, should be „increases”
Line 224: „chelation mechanism released by PGPF siderophores” – mechanism can not be eleased
Line 251-253: The sentence is confusing.
Line 253-255: Hyperparasitism and induction of resistance are quite different processes that should be not included in antagonism. The sentence is misleading.
Line 279: Is „pathogen”, should be „pathogens”
Line 319: Is „thing”, should be „barrier”
Line 319-320: What is a plant system?
Line 366: This sentence is factually incorrect.
Line 368: Please add „such as”
Line 374: What is PPO?
Line 377: Gene expression or expression of PPO activity?
Line 400-415: This part does not describe the biochemical defense
Line 425: The title should be „Defense Signaling” as no molecular markers are described.
Line 512-523: This part does not describe abiotic stress improvement
The tables are not prepared correctly. The columns appear to be shifted relative to each other.
Author Response
RESPONSE TO “REVIEWER 2” COMMENTS
We profusely thank the reviewers for their constructive comments. Herewith we are submitting the revised manuscript following incorporation of all the suggestions as indicated by the reviewers. We hope the reviewers will be happy with the corrections incorporated. In the revised manuscript, the questions raised have been addressed and the changes are made in red colour throughout the manuscript.
Comment No. 1: The subsection, “Antagonism” is unfortunately chaotically written. It presents some examples of different plant-microorganism interaction, but much of them is not an example of antagonism.
Answer: We thank the reviewer for their critical comment and entire “Antagonism” section has been rewritten citing appropriate references in the revised manuscript.
Comment No. 2: In the subsection, describing induction of resistance, authors write about Systemic Acquired Resistance (SAR), but they do not present any examples of this type of resistance.
Answer: As per the suggestion, examples on SAR from fungi have been included and the sub-section has been renamed as “Induction of Resistance” in the revised manuscript.
Comment No. 3: In the subsection, Morphological and Histochemical Defense”, the authors also describe the biochemical response of plants to pathogens, including ROS and phenolic compounds generation. The name of the subtitle should be changed or some information should be transferred to the next subsection.
Answer: As per the suggestion, the information on biochemical and others not related to the subsection “Morphological and Histochemical Defense” has been transferred to the related sections in the revised manuscript.
Comment No. 4: In my opinion, the section„ PGPF as a source of alternatives” which, taking into account the title of the manuscript should be the longest part, is too short.
Answer: As per the suggestion, the section “PGPF as a source of alternatives” has been reframed in the revised manuscript highlighting the role of PGPFs.
Comment No. 5: Authors should use full names or abbreviations consequently throughout the text. Moreover, they should avoid colloquial phrases.
Answer: As per the suggestion, abbreviations have been given in full form at the first citation and abbreviated accordingly in the later portions throughout the revised manuscript. A list of Abbreviations has been included in the revised manuscript.
Comment No. 6: Line 71: Is„ signalling”, should be „signaling”
Answer: As per the suggestion, it is modified in the revised manuscript.
Comment No. 7: Line 101: Is„cytokinine”, should be „cytokinin”
Answer: As per the suggestion, it is corrected in the revised manuscript.
Comment No. 8: Line 120: Is „to augment”, should be „augmenting”
Answer: As per the suggestion, it has been modified in the revised manuscript.
Comment No. 9: Line 132: Is „to increase”, should be „to the increase”
Answer: As per the suggestion, it is corrected in the revised manuscript.
Comment No. 10: Line 149: Is „to increase”, should be „increasing”
Answer: As per the suggestion, it is corrected in the revised manuscript.
Comment No. 11: Line 184 – 192 should be not included in the manuscript
Answer: As per the suggestion, the lines 184 to 192 have been deleted in the revised manuscript.
Comment No. 12: Line 218: What is HCN?
Answer: As per the suggestion, the full form of HCN is provided in the manuscript at the first citation and has been abbreviated thereafter in the revised manuscript.
Comment No. 13: Line 218: Is „that,”, should be „that”
Answer: As per the suggestion, it has been modified accordingly in the revised manuscript.
Comment No. 14: Line 219: Is „increase”, should be „increases”
Answer: As per the suggestion, it has been modified accordingly in the revised manuscript.
Comment No. 15: Line 224: „chelation mechanism released by PGPF siderophores” – mechanism cannot be eleased
Answer: As per the suggestion, the sentence (Line 212-214) has been modified accordingly for better understanding in the revised manuscript.
Comment No. 16: Line 251-253: The sentence is confusing.
Answer: As per the suggestion, the entire “Antagonism” section has been rewritten for better understanding in the revised manuscript
Comment No. 17: Line 253-255: Hyperparasitism and induction of resistance are quite different processes that should be not included in antagonism. The sentence is misleading.
Answer: As per the suggestion, the entire “Antagonism” section has been rewritten for better understanding in the revised manuscript.
Comment No. 18: Line 279: Is „pathogen”, should be „pathogens”
Answer: As per the suggestion, it has been modified accordingly in the revised manuscript.
Comment No. 19: Line 319: Is „thing”, should be „barrier”
Answer: As per the suggestion, it has been modified accordingly in the revised manuscript.
Comment No. 20: Line 319-320: What is a plant system?
Answer: As per the suggestion, the sentence has been modified accordingly in the revised manuscript
Comment No. 21: Line 366: This sentence is factually incorrect.
Answer: As per the suggestion, the sentence (Line 393-395) has been reframed in the revised manuscript.
Comment No. 22: Line 368: Please add „such as”
Answer: As per the suggestion, it has been modified accordingly in the revised manuscript.
Comment No. 23: Line 374: What is PPO?
Answer: As per the suggestion, the full form of PPO is provided in the manuscript at the first citation and has been abbreviated thereafter in the revised manuscript.
Comment No. 24: Line 377: Gene expression or expression of PPO activity?
Answer: As per the suggestion, the sentence has been modified accordingly for better understanding in the revised manuscript.
Comment No. 25: Line 400-415: This part does not describe the biochemical defense
Answer: As per the suggestion, the sentences (Line 400-415) have been replaced to appropriate section in the revised manuscript.
Comment No. 26: Line 425: The title should be „Defense Signaling” as no molecular markers are described.
Answer: As per the suggestion, the title has been modified to “Defense signaling” in the revised manuscript.
Comment No. 27: Line 512-523: This part does not describe abiotic stress improvement
Answer: As per the suggestion, the sentences in Line 512-523 have been removed in the revised manuscript.
Comment No. 28: The tables are not prepared correctly. The columns appear to be shifted relative to each other.
Answer: As per the suggestion, the tables have been presented carefully in order to differentiate between the columns in the revised manuscript.
Round 2
Reviewer 2 Report
After reading the revised manuscript I can see that corrections have been made to the paper. However, some suggestions have not been implemented yet. In connection with the above, I propose one more analysis of the manuscript and the correction of mistakes.
I mark the approved changes in yellow and the recommendations in blue in this response.
Comment No. 1: The subsection, “Antagonism” is unfortunately chaotically written. It presents some examples of different plant-microorganism interaction, but much of them is not an example of antagonism.
Answer: We thank the reviewer for their critical comment and entire “Antagonism” section has been rewritten citing appropriate references in the revised manuscript.
Response: The section has been corrected.
Comment No. 2: In the subsection, describing induction of resistance, authors write about Systemic Acquired Resistance (SAR), but they do not present any examples of this type of resistance.
Answer: As per the suggestion, examples on SAR from fungi have been included and the sub-section has been renamed as “Induction of Resistance” in the revised manuscript.
Response: The section has been corrected.
Line 263: Change “ressitance” to “resistance”.
Line 267: Change “qaquired” to “acquired”
Line 307 and 311: Change “pathogenesis related” to “pathogenesis-related”
Line 312: Remove “the systemic acquired resistance” and leave abbreviation.
Line 316: Change “disease causing” to “disease-causing”
Comment No. 3: In the subsection, Morphological and Histochemical Defense”, the authors also describe the biochemical response of plants to pathogens, including ROS and phenolic compounds generation. The name of the subtitle should be changed or some information should be transferred to the next subsection.
Answer: As per the suggestion, the information on biochemical and others not related to the subsection “Morphological and Histochemical Defense” has been transferred to the related sections in the revised manuscript.
Response: The section has been corrected.
Line 360: Change “superoxide’s” to “superoxide”
Line 361-362: Remove “reactive oxygen species” and leave abbreviation.
Line 391: Remove “also”
Line 396: Change “poly phenoil oxidase (PPO)” to “polyphenol oxidase (PPO)”
Comment No. 4: In my opinion, the section„ PGPF as a source of alternatives” which, taking into account the title of the manuscript should be the longest part, is too short.
Answer: As per the suggestion, the section “PGPF as a source of alternatives” has been reframed in the revised manuscript highlighting the role of PGPFs.
Response: The section has been corrected only partly.
Please add any references in this section.
Moreover, any examples of PGPF as a source of alternatives should be added.
Additional comments:
Line 542: Remove space in “relationships , the”
Line 553: Add “fungicides”
Line 570: Change ”eval;uation” to “evaluation”
List of Abbreviations: Remove the unnecessary line in the table.
Comment No. 5: Authors should use full names or abbreviations consequently throughout the text. Moreover, they should avoid colloquial phrases.
Answer: As per the suggestion, abbreviations have been given in full form at the first citation and abbreviated accordingly in the later portions throughout the revised manuscript. A list of Abbreviations has been included in the revised manuscript.
Response: Please, check one more time whether abbreviations are not explained more than one time, as in the case of SAR or ROS.
Comment No. 6: Line 71: Is„ signalling”, should be „signaling”
Answer: As per the suggestion, it is modified in the revised manuscript.
Line 430: Change “signalling” to “signaling”
Comment No. 7: Line 101: Is„cytokinine”, should be „cytokinin”
Answer: As per the suggestion, it is corrected in the revised manuscript.
Comment No. 8: Line 120: Is „to augment”, should be „augmenting”
Answer: As per the suggestion, it has been modified in the revised manuscript.
Comment No. 9: Line 132: Is „to increase”, should be „to the increase”
Answer: As per the suggestion, it is corrected in the revised manuscript.
Comment No. 10: Line 149: Is „to increase”, should be „increasing”
Answer: As per the suggestion, it is corrected in the revised manuscript.
Comment No. 11: Line 184 – 192 should be not included in the manuscript
Answer: As per the suggestion, the lines 184 to 192 have been deleted in the revised manuscript.
Comment No. 12: Line 218: What is HCN?
Answer: As per the suggestion, the full form of HCN is provided in the manuscript at the first citation and has been abbreviated thereafter in the revised manuscript.
Comment No. 13: Line 218: Is „that,”, should be „that”
Answer: As per the suggestion, it has been modified accordingly in the revised manuscript.
Comment No. 14: Line 219: Is „increase”, should be „increases”
Answer: As per the suggestion, it has been modified accordingly in the revised manuscript.
Comment No. 15: Line 224: „chelation mechanism released by PGPF siderophores” – mechanism cannot be eleased
Answer: As per the suggestion, the sentence (Line 212-214) has been modified accordingly for better understanding in the revised manuscript.
Comment No. 16: Line 251-253: The sentence is confusing.
Answer: As per the suggestion, the entire “Antagonism” section has been rewritten for better understanding in the revised manuscript
Comment No. 18: Line 279: Is „pathogen”, should be „pathogens”
Answer: As per the suggestion, it has been modified accordingly in the revised manuscript. Comment No. 19: Line 319: Is „thing”, should be „barrier”
Answer: As per the suggestion, it has been modified accordingly in the revised manuscript.
Comment No. 20: Line 319-320: What is a plant system?
Answer: As per the suggestion, the sentence has been modified accordingly in the revised manuscript
Comment No. 21: Line 366: This sentence is factually incorrect.
Answer: As per the suggestion, the sentence (Line 393-395) has been reframed in the revised manuscript.
Comment No. 22: Line 368: Please add „such as”
Answer: As per the suggestion, it has been modified accordingly in the revised manuscript.
Comment No. 23: Line 374: What is PPO?
Answer: As per the suggestion, the full form of PPO is provided in the manuscript at the first citation and has been abbreviated thereafter in the revised manuscript.
Comment No. 24: Line 377: Gene expression or expression of PPO activity?
Answer: As per the suggestion, the sentence has been modified accordingly for better understanding in the revised manuscript.
Comment No. 25: Line 400-415: This part does not describe the biochemical defense
Answer: As per the suggestion, the sentences (Line 400-415) have been replaced to appropriate section in the revised manuscript.
Comment No. 26: Line 425: The title should be „Defense Signaling” as no molecular markers are described.
Answer: As per the suggestion, the title has been modified to “Defense signaling” in the revised manuscript.
Comment No. 27: Line 512-523: This part does not describe abiotic stress improvement
Answer: As per the suggestion, the sentences in Line 512-523 have been removed in the revised manuscript.
Comment No. 28: The tables are not prepared correctly. The columns appear to be shifted relative to each other.
Answer: As per the suggestion, the tables have been presented carefully in order to differentiate between the columns in the revised manuscript.
Response: The above mistakes have been corrected.
Comment No. 17: Line 253-255: Hyperparasitism and induction of resistance are quite different processes that should be not included in antagonism. The sentence is misleading.
Answer: As per the suggestion, the entire “Antagonism” section has been rewritten for better understanding in the revised manuscript.
Response: The section has been corrected but still includes hyperparasitism into antagonistic reactions, which is not correct.
Author Response
RESPONSE TO “REVIEWER 2” COMMENTS
We profusely thank the reviewers for their constructive comments. Herewith we are submitting the revised manuscript following incorporation of all the suggestions as indicated by the reviewers. We hope the reviewers will be happy with the corrections incorporated. In the revised manuscript, the questions raised have been addressed and the changes are made in red colour throughout the manuscript.
Comment No. 1: Line 263: Change “ressitance” to “resistance”.
Answer: As per the suggestion, it has been modified accordingly in the revised manuscript.
Comment No. 2: Line 267: Change “qaquired” to “acquired”
Answer: As per the suggestion, it has been modified accordingly in the revised manuscript.
Comment No. 3: Line 307 and 311: Change “pathogenesis related” to “pathogenesis-related”
Answer: As per the suggestion, it has been modified accordingly in the revised manuscript.
Comment No. 4: Line 312: Remove “the systemic acquired resistance” and leave abbreviation.
Answer: As per the suggestion, it has been modified accordingly in the revised manuscript.
Comment No. 5: Line 316: Change “disease causing” to “disease-causing”
Answer: As per the suggestion, it has been modified accordingly in the revised manuscript.
Comment No. 6: Line 360: Change “superoxide’s” to “superoxide”
Answer: As per the suggestion, it has been modified accordingly in the revised manuscript.
Comment No. 7: Line 361-362: Remove “reactive oxygen species” and leave abbreviation.
Answer: As per the suggestion, it has been modified accordingly in the revised manuscript.
Comment No. 8: Line 391: Remove “also”
Answer: As per the suggestion, it has been modified accordingly in the revised manuscript.
Comment No. 9: Line 396: Change “poly phenoil oxidase (PPO)” to “polyphenol oxidase (PPO)”
Answer: As per the suggestion, it has been modified accordingly in the revised manuscript.
Comment No. 10: Please add any references in this section. Moreover, any examples of PGPF as a source of alternatives should be added.
Answer: As per the reviewer suggestion, references have been added in the sub-section “PGPF as a source of alternatives” in the revised manuscript.
Comment No. 11: Line 542: Remove space in “relationships , the”
Answer: As per the suggestion, it has been modified accordingly in the revised manuscript.
Comment No. 12: Line 553: Add “fungicides”
Answer: As per the suggestion, it has been modified accordingly in the revised manuscript.
Comment No. 13: Line 570: Change”eval;uation” to “evaluation”
Answer: As per the suggestion, it has been modified accordingly in the revised manuscript.
Comment No. 14: List of Abbreviations: Remove the unnecessary line in the table.
Answer: As per the suggestion, it has been modified accordingly in the revised manuscript.
Comment No. 15: Please, check one more time whether abbreviations are not explained more than one time, as in the case of SAR or ROS.
Answer: As per the reviewer suggestion, the entire manuscript has been re-checked thoroughly for their appropriateness.
Comment No. 16: Line 430: Change “signalling” to “signaling”
Answer: As per the suggestion, it has been modified accordingly in the revised manuscript.
Comment No. 17: The section has been corrected but still includes hyperparasitism into antagonistic reactions, which is not correct.
Answer: As per the reviewer suggestion, the section “Antagonism” has been relooked and the details of “hyperparasitism” have been removed in the revised manuscript. We further state here that, all the references quoted in the section “Antagonism” has been checked thoroughly to authenticate the antagonistic nature.

Round 3
Reviewer 2 Report
Unfortunately, Authors have not corrected the mistakes which I suggested. The revised manuscript is the same as after the first revision. Therefore I recommend it again to the major revision according to my comments from the review2.
Author Response
Comment 1: Unfortunately, Authors have not corrected the mistakes which I suggested. The revised manuscript is the same as after the first revision. Therefore I recommend it again to the major revision according to my comments from the review2.
Answer: Here, we would like to state that, we didn’t receive any additional reviewer’s comments apart from first revision. But, we have addressed all the comments that were raised in the second revision in the current revised manuscript as per the suggestion.
Round 4
Reviewer 2 Report
All my suggestions were taken into account. Therefore, I recommend the manuscript for publication in the Journal of Fungi.